# Fatty Acid Profiles and Production in Marine Phytoplankton

**DOI:** 10.3390/md17030151

**Published:** 2019-03-04

**Authors:** Sigrún Huld Jónasdóttir

**Affiliations:** Section for Oceans and Arctic, Technical University of Denmark, Kemitorvet, Building 202, DK-2800 Kgs. Lyngby, Denmark; sjo@aqua.dtu.dk

**Keywords:** marine phytoplankton, fatty acids, fatty acid synthesis, PUFA synthesis, EPA, DHA, SDA, environmental effects

## Abstract

Microalgae are the primary producers of carbon in marine ecosystems, fixing carbon and subsequently generating various biomolecules such as carbohydrates, proteins and lipids. Most importantly, microalgae are the generators and main suppliers of ω3 polyunsaturated fatty acids (ω3PUFA) in the marine ecosystem, which have a fundamental importance for the functioning and quality of the whole marine food web. A meta-analysis of over 160 fatty acid profiles of 7 marine phytoplankton phyla reveals not only a phyla-specific, but also a highly class-specific PUFA production of marine phytoplankton. The highest EPA (Eicosapentaenoic acid; 20:5ω3) production per total fatty acids was found in 2 classes of Haptophyta and in Ochrophyta, while Dinophyta and the Haptophyte *Emiliana huxleyi* show the highest production of DHA (Docosahexaenoic acid; 22:6ω3). An important precursor for EPA, Stearidonic acid (SDA, 18:4ω3) is found in high proportions in Cryptophyta and the Chlorophta class Pyramimonadophyceae. Per unit of carbon, Chlorophyta and Cyanobacteria were the poorest producers of highly unsaturated fatty acids (HUFA). The remaining phyla had a similar HUFA contribution per unit of carbon but with different compositions. The nutritional and environmental effects on the phytoplankton PUFA production is summarized and shows a lowering of the PUFA content under stressful environmental conditions.

## 1. Introduction

The basis of the marine pelagic ecosystem lies with the primary producers, the unicellular phytoplankton that fix inorganic carbon (CO_2_) with the aid of sunlight (photosynthesis). The carbon, fixed in the form of glucose within the phytoplankton, is directed into various types of molecular components mainly combined with phosphorus and/or nitrogen, making up the building blocks of the phytoplankton cell. These building blocks are protein, lipids and carbohydrates, and they are the nutritional foundation for the upper trophic levels in the marine environment, starting with the secondary producers, e.g., copepods and krill, controlling their growth, reproduction, fitness and survival. The ability of zooplankton to concentrate and store phytoplankton-based lipids [1,2,3] means that they are key trophic vectors, channeling these essential nutritional compounds towards fish, seabirds, marine mammals and eventually humans.

This review focuses on the chemical composition of marine phytoplankton as a nutritional source for marine zooplankton that could limit their reproduction and survival and therefore affect the efficiency of the entire marine food web. The starting point is the notion, taken from various studies, that different phytoplankton types offered as food for zooplankton differ greatly in their ability to support zooplankton growth and survival, ranging from being essentially non-nutritious to being excellent food [4,5,6,7]. This suggested some important differences in phytoplankton chemistry that mattered for the nutritional needs of their consumers.

Therefore, it was important to become familiar with the chemical composition of phytoplankton and establish which components determine the quality of the phytoplankton as food for their consumers. The nutritional components are usually the micro-molecules: the building blocks of the different macromolecules. Those building blocks could be, for example, the specific fatty acids, vitamins, trace metals and specific amino acids.

The essential nutrients are the ones that the organisms cannot synthesize themselves, which they have to obtain from their diet. Studies had shown that crustaceans do not or cannot easily biosynthesize the ω3 and ω6 polyunsaturated fatty acids (PUFAs), and that these fatty acids are found in crustaceans in proportion to their availability in their diet [1,3,8,9,10]. Therefore, the focus has been on the fatty acid composition of marine phytoplankton which has indeed shown the importance of PUFA on reproduction and the growth of secondary producers of the oceans [11,12,13,14,15].

Here, I begin by giving a general summary of the chemical composition of marine phytoplankton phyla, with a special emphasis on fatty acid synthesis and biochemistry. I will present the results of a meta-analysis of fatty acid profiles in various phytoplankton groups and summarize which environmental parameters affect the biochemical pathways of fatty acid synthesis.

## 2. The Gross Chemical Composition of Phytoplankton

Carbon is the main element in most molecular structures in the phytoplankton cell and is often used as an indicator of the phytoplankton biomass. However, the quality of carbon can vary greatly based on the compound to which it is bound (Figure 1). In the phytoplankton cell, carbon is found in all macromolecules such as carbohydrates and lipids. Nitrogen is primarily bound in protein, and, as it is essential for phytoplankton growth, the nitrogen content has often been used to indicate the quality (i.e., the nutritional value) of the cell (most often as the C/N ratio). Nitrogen is also an essential part of vitamins, enzymes and some lipid complexes (Figure 1, Figure A1).

Protein is the main organic group measuring ca. 40–60% of the organic mass, with carbohydrates contributing approximately 17–40% and lipids about 16–26% (Figure 2, and references therein). This proportion is, however, dependent on the phytoplankton growth condition, as discussed in Section 3. In general, the average ratio of the protein, carbohydrates and lipids reported is remarkably similar between the different phytoplankton phyla, at approximately 5:3:2 (Figure 2). Proteins are engaged in almost all the tasks of the cellular activities. They are large complex molecules composed of one or more long chains of amino acids. They are important components of all membranes, involved in the transport of other molecules and ions across the membranes. Proteins are as diverse as the functions they serve. Most enzymes are proteins that organize, construct and receive signals, while the structural proteins maintain the shape of the cell. In animals, the structural proteins are the muscles and connective tissues. There are about 20 different types of amino acids that build up the protein structures, 10 of which are essential, (essential amino acids, EAA) i.e., cannot be synthesized de-novo by most organisms. The quality of marine proteins is measured by the presence of these EAAs, some of which have been found to be essential for the growth of some zooplankton species [16,17]. The synthesis of marine EAAs is found to be slower and more susceptible to nitrogen limitation compared to nonessential amino acids [18]. However, the protein quality (EAA) is found to be similar between many phytoplankton species [19], and the differences in protein quality appears to be secondary; that is, it is first evident when the other essential nutrients, such as essential fatty acids, are sufficient in the diet [19].

Carbohydrates are either energy or structural compounds. Sugars are easily mobilized for energy (labile). Starch and glycogen are built up of longer branched polymers and are used for storage. Cellulose and chitin are structural and resistant to digestion (refractory). Marine phytoplankton carbohydrates are mainly glucose, galactose and mannose [19]. From a nutritional point of view, the refractory carbohydrates will not be of high nutritional value for zooplankton with a simple gut, while the labile carbohydrates would provide a more easily mobilized energy, without being of high nutritional value; they contain little in the way of nitrogen, phosphate or other trace elements.

Lipids comprise a wide range of compounds used in a wide variety of functions, such as energy storage, digestion, membrane structure, photosensitive pigments and more. The main lipid types in the phytoplankton cell are triacylglycerol (TAG), galactolipids (GL) and phospholipids (PL) (see Appendix B). The building blocks of these lipid types are fatty acids which are hydrocarbon chains with a carboxyl end (-COOH) in the head of the molecule. The fatty acid chain can be saturated, that is, without a double bond (saturated fatty acid; SAFA), containing one double bond (monounsaturated; MUFA), or with 2 or more double bonds (polyunsaturated; PUFA, Appendix B
Figure A1).

Given the general similarity of the Protein/Carbohydrate/Lipid ratio (5/3/2) in the 7 phytoplankton phyla (Figure 2), these macromolecules cannot explain the variation observed in the consumer’s growth when fed different phytoplankton types. Therefore, it is necessary to look further into the more detailed structure of the classes, and as mentioned before, into the building blocks of lipids: the fatty acids.

### 2.1. From Glucose to Fatty Acids

To get from glucose, produced by photosynthesis, to specific fatty acids and storage lipids, a series of complicated biochemical pathways are required. Knowing these pathways gives a better understanding of what controls and limits the fatty acid makeup of the phytoplankton cell (see references in the caption of Figure 3). A more detailed outline of the specific pathways and location of fatty acid synthesis in a eukaryotic organism can, for instance, be found in Zulu et al. [24] and Mühlroth et al. [25].

Fatty acid synthesis in the algal cell occurs through an aerobic pathway [26] and takes place in the chloroplast and the endoplasmic reticulum. The glucose produced by photosynthesis is converted by glycolysis to pyruvate, which is the molecular basis for all metabolisms. Pyruvate is subjected to oxidative decarboxylation with the coenzyme A (CoASH) to form acetyl-CoA, which is taken into several directions; to the Krebs cycle in the cytosol or to form malonyl-CoA with biotin (acetyl-CoA carboxylase) driven by adenosine triphosphate (ATP). The acetyl-CoA may derive from either the chloroplast itself or from the cytosol [25]. This is the start of the fatty acid synthesis outlined in Figure 3. Fatty acid synthesis can be divided into several steps. Step 1: Fatty acid synthesis in the chloroplast where Malonyl-CoA and Acetyl-CoA contribute 2 carbons each to form the first fatty acid chain (4:0-ACP, Acyl Carrier Protein). Step 2: Associated fatty acid elongation where the 4:0-ACP is successively elongated with the aid of fatty acid synthesase 2 carbons at the time. The cycle ends up with 14–18 carbon length ACP-chains and either enters the fatty acid pool in the cytosol or is taken to step 3. The rest of the fatty acid production involves elongation and desaturation enzymes which occur in the endoplasmic reticulum (ER) in eukaryotic algae [25]. Step 3 involves the first unsaturation step, where 16 or 18:0-ACP with the help of Δ9 desaturase puts the first double bond on the 9th carbon from the ACP end of the chain. The first unsaturated ACP-chain enters the free fatty acid pool of the cell or is further desaturated. Step 4: The desaturation and elongation process takes place in the ER. It takes the 18:1ω9 (or 16:1ω7)-ACP by Δ12 desaturase to form 18:2ω6 further into either the ω6 pathway by Δ6 desaturase or into the ω3 pathway by Δ15 desaturase to form 18:3ω3. Different organisms utilize different desaturases to come to the different PUFAs, but Figure 3 shows two of several possible paths. Step 5: The final step in the lipid synthesis is the formation of storage lipids, usually triacylglycerols (TAG), phospholipids (PL) and galactolipids (GL), that occur in the smooth endoplasmic reticulum (ER) [25]. All require glucose-6-phosphate, to which a saturated fatty acyl-CoA is added with the help of acyl transferase. The second acyl-CoA is added, and phosphatidic acid (PA) is synthesized. Phosphatidic acid is the precursor to several lipid complexes, but diacylgcycerol (DAG), which is a precursor for TAG, PL and GL, is formed by hydrolysis of the phosphate group from phosphatidic acid.

Autotrophs (such as phytoplankton) are the only organisms in the marine environment that can produce linoleic (LA = 18:2ω6) and α-linolenic acid (ALA = 18:3ω3) de-novo from 18:0-ACP [18], but these are precursors for longer chain PUFAs (Figure 3). The reason for this is that higher organisms do not have the required Δ12- and Δ15- desaturase enzymes needed to synthesize LA and ALA from 18:0. A further elongation of LA and ALA to longer chain PUFAs is not easily completed by higher organisms such as the calanoid copepods (the main grazers of microplankton), but if done, they cannot biosynthesize these types of fatty acids with high enough efficiency to meet their growth requirements [27,28], except in some cases [28,29,30,31]. In mammals, such elongation is very slow and is limited by the availability of Delta-6-desaturase [32]. Therefore, phytoplankton are the major source of most PUFAs for most higher consumers [33], and LA, ALA and the longer chain length derivatives produced in the phytoplankton are, from a nutritional point of view, considered to be essential for most higher organisms.

### 2.2. Specific Lipid Content in the Phytoplankton

The different phytoplankton phyla contain different proportions of the lipid types hydrocarbon (HC), triacylglycerol (TAG), free fatty acids (FFA), sterols (ST), pigments, and polar lipids (POL), that include mainly phospholipids (PL) but also galactolipids (GL). Figure 4 depicts the average lipid class composition of 7 of the main phytoplankton phyla as % of the total lipids. The major lipid class in all the phyla are the polar lipids ranging from 40–95% of the total lipids (references in the figure’s caption).

Triacylglycerol contributes up to 30%, free fatty acids up to 10% and sterol about 5% of the lipid pool (see Figure caption for references). Green algae (chlorophyta) have a very low neutral lipid content. Dinoflagellates (dinophyta) have the highest proportion of neutral lipids out of the 7 phyla. It should however be noted that the ratio between these lipid classes varies greatly with the growth condition of the cells (see Section 3).

#### 2.2.1. Fatty Acid Profiles

The studies that report on marine and freshwater phytoplankton fatty acid profiles run in the hundreds, and are composed of thousands of fatty acid profiles (see [50,51]). The meta-analysis presented here is based on 38 publications that report fatty acid profiles of marine phytoplankton. These consisted of over 160 fatty acid profiles from 7 phytoplankton phyla (cited in the caption of Figure 5). Each species profile covered in this meta-analysis is listed in the Appendix A with additional phyla, class, and genus averages. Here, I identify specific aspects of the analyses, and draw up the apparent differences between classes or orders within a phylum (Figure 5). Figure A2 in the Appendix B depicts the combined average fatty acid profile for the groups. While sch comparisons are a rather tedious read, it is important to understand the differences between phyla when using the fatty acids as biomarkers and tracers (e.g., [3,52,53,54]), when delving into the details of food quality [55,56], or when searching for a potential nutra- or pharmaceutical source for culturing.

By looking at the proportion of fatty acid unsaturation in the different groups, (Figure 5A) it is evident that the highest proportion of PUFA is in Chloro- and Cryptophyta, with about 60% of the total fatty acids. The lowest PUFA is found in Ochrophyta, Cyanobacteria (blue-green algae) and diatoms (22%, 26% and 28% respectively). The details of the differences is apparent in the other figure panels. Panel 5B shows the C16 fatty acids, and panel 5C lists the C18 fatty acids excluding 18:5ω3 (Octadecapentaenoic acid, OPA). Panel 5D contains the nutritionally most important PUFA, the highly unsaturated fatty acids (HUFA) 20:5ω3 (Eicosapentaenoic acid; EPA), 22:6ω3 (Docosahexaenoic acid; DHA) and 20:4ω6 (Arachidonic Acid; ARA), including the shorter 18:5ω3 fatty acid. I include OPA with EPA, DHA and ARA as it appears to be exclusive with EPA, i.e., it may appear in many instances that the phytoplankton synthesize either OPA or EPA, but rarely both.

For ease of reading, the classes and orders are identified by their ID letter, as shown in Figure 5 and in the Appendix A.

Chlorophyta (green algae) contain 12–28% of their fatty acids as C16 fatty acids. They are split in 2 distinctive groups: B & D with 16:2 and 16:3 fatty acids, and the other classes (A, C, E & F) with 16:4 fatty acids. The C18 fatty acids make up 35–43% of the total fatty acids of green algae, and all groups have a high proportion of 18:3 ω3 fatty acids. The remaining C18 types split the classes into another combination of groups: E & F, having mainly 18:1ω7 and stearidonic acids 18:4ω3 (SDA); and the others (A, B, C & D), who have 18:1ω9 and 18:2ω6. A third combination between the green algal classes is apparent in the HUFA fraction, where A, B, C & D contain mainly EPA (10–25%) and E & F, mainly OPA and DHA (12%).

The C16 fatty acid group is a signature for silica rich Diatoms (>40%), especially the 16:1ω7, while having low levels of C18 fatty acids. The only difference between the two diatom classes shown here is in their 16:3 PUFA content, where 16:3ω3 and 16:3ω6 are present in class G but not in class H. Over 10% of the total diatom fatty acids are in the long chain EPA.

In contrast, Dinophyta (dinoflagellates) are high (20%) in C18 fatty acids and especially >20PUFA, where 22:6ω3 (DHA) and 18:5ω3 are the signature fatty acids for the phyla. One of the orders, J, has a low contribution of 18:5ω3 fatty acids but higher 20:5ω3.

The fatty acid profiles of the three orders of Haptophyta that are listed differ from each other in their C16, C18 and >20PUFA composition. Even within the class Coccolithophyceae, there are distinct differences, mainly because of a special composition of *E. huxleyi* diverging from the others within the class, which is therefore presented by itself. *E. huxleyi* (O) has a lower HUFA content, and a higher ω6 content compared to the rest of the class, with 18:3ω3 and a high proportion of DHA. The Pavlovo- and Chrysophyceae have similar profile and are combined as group Q. They differ from the other Haptophyta containing, high proportion of 20:5ω3 and 16:1ω7. The orders within Cyanobacteria have generally similar profiles, totally lacking the long chain PUFA. The 18:3ω3 fatty acid is conspicuous in all classes, and 16:1ω9 is about 20–30% of the total fatty acids in orders R and S, which also contain 18:3ω6, while the orders T&U have 16:1ω7 and lack 18:3ω6. The fatty acid composition and dynamics of marine Cryptophyta has recently been covered in more detail than here in [89]. Generally the Cryptophyta are low in C16 fatty acids but have equal mixtures of all 18:3ω3 and 18:4ω3 fatty acids, though they lack 18:5ω3 fatty acids. Both EPA and DHA are well represented within the profile. The Ochrophyta has a similar profile to diatoms with a high proportion of 16:1ω7 and EPA, but additionally has 16:1ω9 fatty acids and a total lack of DHA; however, it has Arachidonic acid (ARA, 20:4ω6), which is not present or reported in noticeable amounts in other phyla (See Figure 6).

#### 2.2.2. PUFA as a Fraction of Biomass

All the fatty acid profiles above are presented as % of the total fatty acid. However, what matters for a consumer is the actual amount of fatty acid in the food, or how much quality it receives per carbon (or dry weight) ingested. Very few of the studies with the fatty acid profiles give the specifics of the phytoplankton analyzed, such as the size, carbon or total fatty acids. If the size is given, it is usually possible to calculate the carbon content [90], while the total lipids or fatty acids per cell are seldom given. Table 1 summarizes several carbon-based specifics of the fatty acids available from the meta-analysis literature. The scarcity of data is reflected in the variation of the mean. The fatty acids in the different phytoplankton groups range between 5–14% of the carbon biomass, with the lowest percent occurring in dinoflagellates and Chlorophyta. Despite the relatively high PUFA content, the proportion of HUFA (including 18:5ω3) generates the difference between the phyla with Chlorophyta having the lowest proportion of HUFA. Other indicators of quality are the ω3/ω6 [82,87,91] and DHA/EPA ratios [55,92,93], both of which are high in diatoms.

## 3. Environmental Effects on Lipid and Fatty Acid Composition

In the 1980s, Mayzaud et al. and Morris et al. [94,95] reported a strong seasonality in the carbon, nitrogen, protein, carbohydrate and lipid content of seston in nature. Additionally, both laboratory and field studies have shown that phytoplankton undergo compositional changes in their lipid classes and specific fatty acids as nutrient availability changes [55,87,94,96,97,98]; these compositional changes are also related to the age of the phytoplankton culture [57,60,99]. It was therefore inevitable that those changes would affect copepod egg production rates and growth, especially if those changes affect the essential nutritional components such as the ω3 fatty acids. A series of studies have focused on changes in the fatty acid content of phytoplankton, with a focus on change in physical condition, as well as nutritional and metal limitations.

### 3.1. Physical Environment

The temperature is one of the main environmental factors that can influence the biology of organisms. Each species (and even strain) has its own window of optimal growth and metabolic function, and a small change in temperature can change the dominance of phytoplankton species in an ecosystem. For lipid synthesis, temperature has been shown to affect the formation of the RuBisCO enzyme, which is a key factor in the carbon assimilation in algae and thus the ability to produce glucose – the precursor of fatty acid synthesis (Figure 3). Generally, studies show an increased lipid production, mainly TAG with increased temperatures. This is mainly shown as an increase in the SAFA and MUFA production. The saturation index of some GL has been shown to decrease with higher temperatures in a freshwater dinoflagellate. (Based on: [38,82,100,101,102,103]).

Light is needed for the generation of Nicotinamide adenine dinucleotide phosphate (NADPH) and acetyl-CoA carboxylase, both of which are essential for fatty acid synthesis. Elongation and desaturation of ARA to EPA and to DHA has been shown to be light-dependent in the haptophyte *Pavlova lutheri* (group Q), where HUFA production is active under low light conditions [104]. Other studies testing different light levels indicate that lipid production depends on the growth stage of the cultures and species, and that it appears to depend on the type of sugar used as an energy source to fuel the fatty acid production. (Based on: [19,38,100,105,106].

### 3.2. Nutrients

Many of the lipid measurements that represent nutrient limitations were conducted on cultures in a stationary growth phase, so the results are a combination of nutrient and light limitations.

Nitrogen is an essential part of amino acid synthesis, and when limited, the path is shifted towards non-nitrogenous compounds such as lipid or carbohydrate synthesis. When nutrient limitation becomes critical, it causes the size of thylakoids and other cell membranes to decrease, affecting the absolute amounts of PUFA, until the limitation becomes critical and affects the turnover of enzymes and the ability to repair or synthesize membranes; this results in the recycling PL and GL and the associated PUFA. The total sterol content of the cell (in diatoms and chlorophytes) has been shown to decrease with nitrogen stress. (Based on: [80,107,108,109]).

Studies on phosphate limitation show that SAFA and MUFA increase at the cost of PUFA. Total lipids increased in diatoms and prymnesiophytes, while total lipids decreased in chlorophytes—since chlorophytes store carbon as carbohydrates but not as lipids, and the P limitation affects their ability to synthesize phosphoglycerolipids. TAG and galactolipid content increased at the cost of phospholipids, indicating that the phosphate limitation pushes synthesis towards TAG and sugars. (Based on: [48,109,110,111]).

Silica limitation acts on diatoms, but mainly on their division rates. The lipid content, especially TAG, is found to increase with the Si limitation, and SAFA and MUFA are found in a higher proportion in the phytoplankton cell, compared to Si replete cultures. The lipid increase has been found to be equal to expectations in the 2 daughter cells as the frustule formation halts when Si is limited, while other processes continue causing an increase in the lipid storage. (Based on: [105,107,112,113]).

The reason why the PUFA and ω3 production is apparently affected by the N and P limitation is not clear, but it is most likely related to the need for NADP during desaturation and elongation.

### 3.3. Trace Metals

The trace metals essential for phytoplankton growth include manganese (Mn), Iron (Fe), cobalt (Co), copper (Cu), zinc (Zn) and nickel (Ni). Not many studies have been conducted to investigate the effects of trace metal limitation on fatty acid composition.

An increase in manganese (Mn^+2^) availability has been shown to cause an increase in PUFA in autotrophs [100]. Manganese is important in photosynthesis and has been shown to limit the chlorophyll content of cells [114].

Iron (Fe) is an important trace element used in the photosynthetic electron transport as Fe_2_S_2,_ and it acts as an electron donor for the production of NATP. As mentioned above, lipid production is energy-dependent and requires, for example, 14 NADPH and 7 ATP for the production of one mole of palmitate (16:0). The limitation of iron results in a reduction of phytoplankton cell volumes by half and a significantly lower total lipid content in cells. The production of SDA is hindered in Fe-depleted cells compared to Fe-replete cells [87], which can be traced to the importance of Fe in the composition of the fatty acid desaturases (see above). Fe forms a reactive complex with oxygen (diiron) in the desaturation molecule, but oxygen reacts with carbon in the fatty acid chain and converts single bonds to double (From: [86,115,116]).

Generally, the different factors listed above limit the pathways shown in Figure 3 at different or various levels. Many are essential in the photosynthetic pathway where light activates the Mn+2 (and Mg+) dependent chlorophyll molecule (also containing nitrogen) and temperature affects the carbon fixation rates (RuBisCO), along with other rates. Iron is essential in the electron transport chain as an electron donor in the NADP formation. Nitrogen limits the amino acid production, as well as being an essential part of most enzymes, NADP and phospholipids. Phosphate is also essential in the energy transfer of ADP and ATP.

All these factors control phytoplankton growth and chemical composition in nature and can certainly be used to manipulate phytoplankton in cultures, for example by changing the light availability (density of cultures) and nutrients to attain the required and desired lipid and fatty acid composition.

It should be noted that most of the listed differences are most often relative (percentages) and do not reflect the absolute changes in the fatty acid composition. However, while the environmental and nutritional factors affect the relative fatty acid composition, the specificities of the fatty acid signature of the different phytoplankton phyla are relatively stable, and statistical analyses by Galloway and Winder [50] show that phylogenic fatty acid signatures are more robust than some fatty acid shifts that happen due to environmental factors. Therefore, using fatty acids as biomarkers is still a robust tool not greatly affected by environmental changes. However, the absolute amount of essential fatty acids is of crucial importance for the food web dynamics, and shifts in the absolute value of EFA will affect the quality of the phytoplankton as food.

## 4. Discussion

Lipids are immensely important for the functioning and well-being of marine ecosystems. Essential omega-3 lipids are produced by phytoplankton, and accumulated and transferred by zooplankton through the entire marine food web, part of which eventually ends up on our dinner tables. The quality, efficiency and productivity of the marine food web is highly dependent on the type of primary producer dominating at every moment, as is underscored by the great variation in the essential fatty acid content of the different phytoplankton phyla and classes.

Marine lipids are in huge demand [117,118] and have a high economic value [119]. The industrial uses of marine lipids are related to human consumption, fisheries, aquaculture, agriculture, health and cosmetics. Omega-3 PUFA are essential for the development and function of the brain, the nervous system and eyes, as well as serving as a preventative for heart disease and inflammation [120]. Therefore, EPA and DHA in particular are highly sought after by the nutra- and pharmaceutical industry.

Microalgae are an excellent source for acquiring ω3 and ω6 PUFA. The meta-analysis clearly reveals that some phytoplankton classes are more suitable sources for essential PUFA than others. For cultivation purposes, it is important to be aware that while the average lipid content of all phytoplankton phyla is similar (about 20% of their organic matter content), the lipid type differs greatly between phyla (Figure 4), as do the types and proportion of the ω3 and ω6 fatty acids. In the literature, fatty acid profiles are usually presented as a fraction of the total fatty acid pool. However, when looking at HUFA as a fraction of the biomass (Figure 6), it is clear that some phytoplankton types give more EPA and DHA per unit of carbon than others. The average PUFA content of the different phyla ranges from 0 (Cyanobacteria) to 2.5% (Ochrophyta) of the carbon biomass. Of these, the contribution of EPA and DHA to the PUFA mass varies both in proportion and in amount.

In the absence of EPA and DHA, SDA might be another PUFA of interest with human health benefits. SDA is synthesized from alpha-linolenic acid (18:3n3, ALA) with the aid of delta-6 desaturase, and as such is a precursor of EPA and DHA. Delta-6-desaturase is a limiting enzyme in humans [32] and is thought to decline in humans with age [120]. Delta-6-desaturase has several potential functions in the lipid desaturation pathway (Figure 3) that could compete for the generation of ω3 versus ω6 PUFA. In particular, the phytoplankton groups with a high contribution of SDA are Cryptophyta and the class Pyramionadophycea within the Chlorophyta.

It can be argued that, globally speaking, the marine ecosystem is in a state of transition. The Arctic and subarctic North Atlantic are, in particular, transitioning to warmer waters and decreased salinity due to the melting of sea ice and the influx of glacial melt waters [121]. These systems are predominantly fuelled by the diatom vernal bloom that are highly characteristic for seasonal environments (e.g., polar and subpolar seas) and which make high quality EPA available for the marine food web. The warming and freshening of the subarctic waters is predicted to cause an increased stratification of the water masses, that will limit the nutrient input from deeper water masses causing nutrient limitation in the systems, and disrupting their highly productive seasonal cycle. Based on the studies listed above, both the increased temperatures and nutrient limitation decrease the quality of lipids in phytoplankton; while the total lipid content increases, the PUFA fraction goes down. At this stage, the question of what will happen with microplankton diversity and which organisms may take over from diatoms remains speculative, but it will most likely lessen the quality and efficiency of the Arctic food web. This is of great concern, and might even increase the need for additional production chains of PUFA in the near future, to fulfil the demand for this essential nutritional component for human consumption and health.

## 5. Methods

All metadata were from publications on marine phytoplankton, and freshwater species were not included in the analyses. The species were sorted according to phyla, class order and genus using the criteria from Algae Base [122]. The gross chemical composition of phytoplankton in the literature is usually presented as % dry weight or % organic weight. To facilitate comparisons, the data is presented here as the relative proportion between the 3 organic groups; protein, carbohydrates and lipids.

The presentation of fatty acid data is as a % of the total fatty acids. After an inspection of the profile similarities on the species and genus level, averages were taken on a class-level for most of the phyla. For Dinophycea, the orders were very different within the classes, and their profiles and are here presented on an order-level. There is some discrepancy between publications in the totality of the fatty acid profiles presented—e.g., some studies do not identify all fatty acids, and ω6 fatty acids may be underrepresented in some studies. It is not possible to evaluate if in those instances the specific fatty acids are not present in the sample or not analyzed/recognized. Therefore, the averages are based on >0.1% presence of the specific fatty acids, and zeros are not included in the averages. The original data is presented as Appendix A.

## Figures and Tables

**Figure 1 marinedrugs-17-00151-f001:**
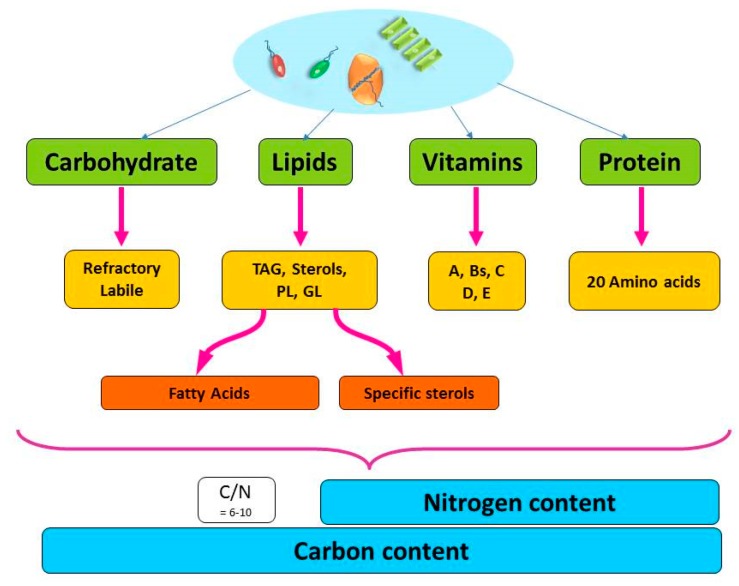
Schematic diagram of the main macro-molecular structures of the phytoplankton cell and further fractionation of those macromolecules to the building blocks that make up the particulate carbon and nitrogen pool. TAG: Triacylglycerol, PL: Phospholipids, GL: Galactolipids.

**Figure 2 marinedrugs-17-00151-f002:**
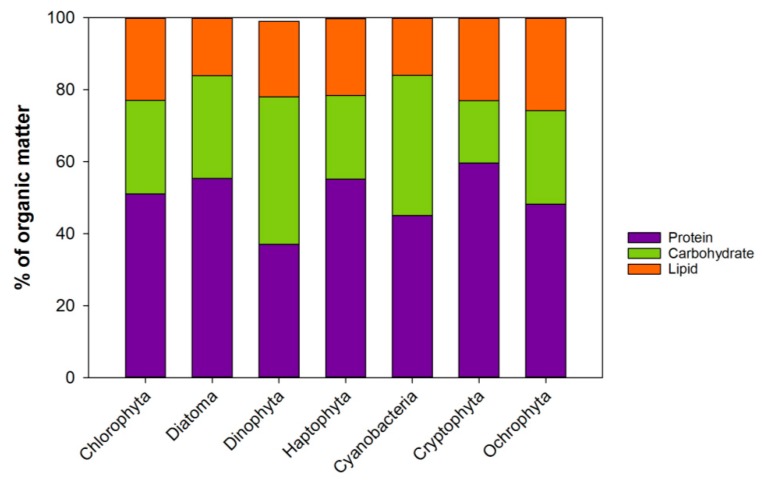
The average biochemical composition of 7 phytoplankton phyla. Compilation of data from: [19,20,21,22]. Updated from [23].

**Figure 3 marinedrugs-17-00151-f003:**
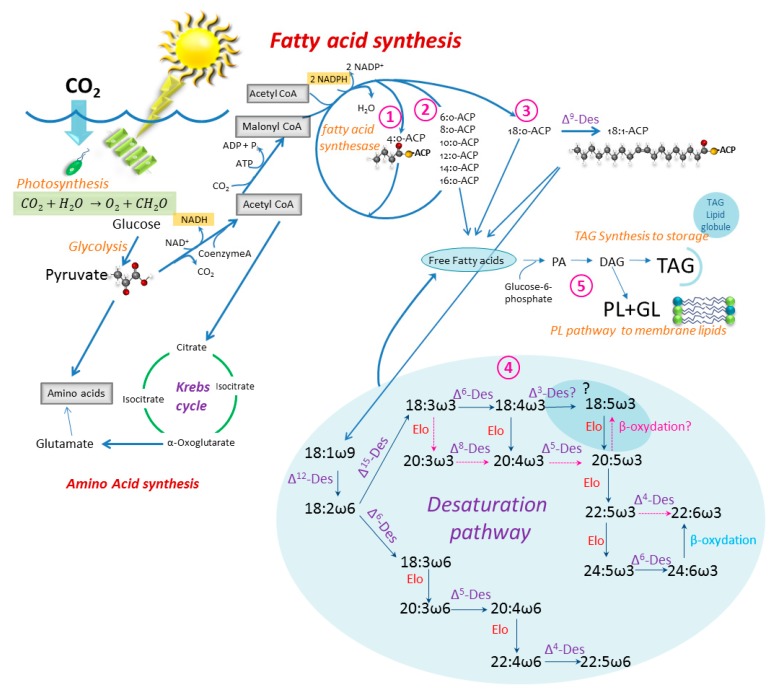
Simplified sketch of the fatty acid synthesis in the phytoplankton cell; from photosynthesis (glucose) to triacylglycerol (TAG), phospholipid (PL) and galactolipid (GL). The path towards amino acid synthesis is shown. Des: Desaturase, Elo: elongase, ACP: Acyl carrier protein. Inspired by: [25,33,34,35,36,37,38]. The desaturation pathway from [39,40], with an alternative pathway to 18:5ω3 as suggested by [41], is shown by red arrows. Paths ①–⑤ are discussed in the text. Updated from [23].

**Figure 4 marinedrugs-17-00151-f004:**
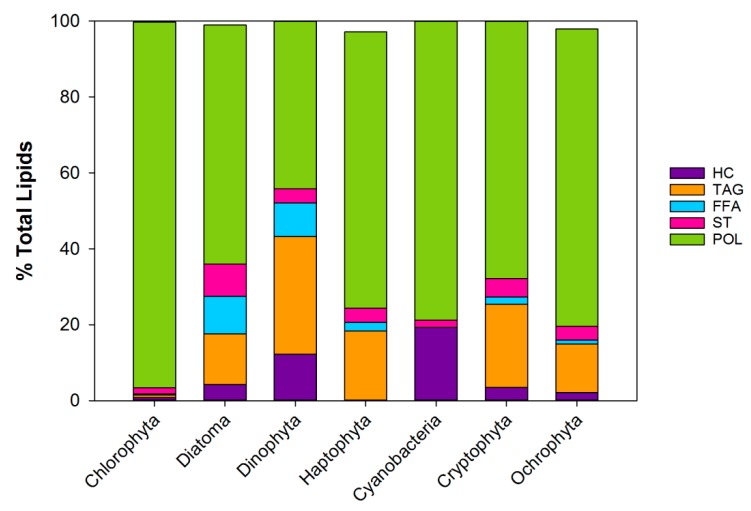
Lipid class composition of 7 phyla of phytoplankton as % of the total lipids excluding pigments. Based on a compilation of 9 articles [1,42,43,44,45,46,47,48,49]. HC: Hydrocarbon, TAG: Triacylglycerol, FFA: Free fatty acids, ST:sterol, POL:polar lipids (PL + GL). Updated from [23].

**Figure 5 marinedrugs-17-00151-f005:**
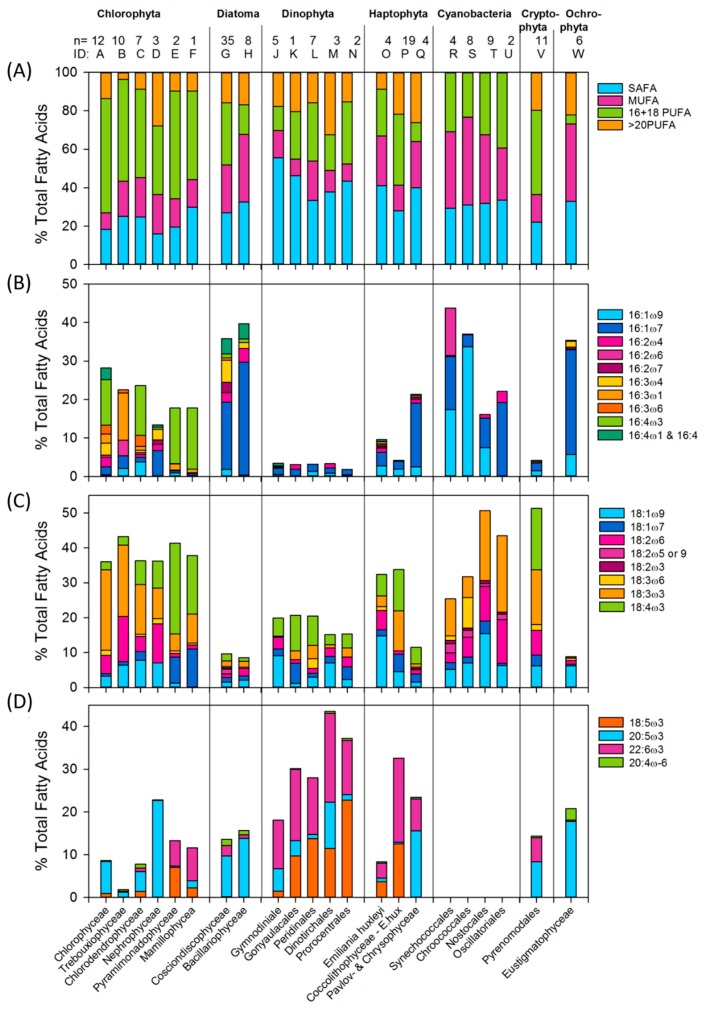
Fatty acid profiles of 7 phytoplankton phyla and 22 classes/orders as % of the total fatty acids. n: number of profiles behind the analysis. The proportion of (**A**) Saturated fatty acids (SAFA), mono-unsaturated fatty acids (MUFA), 16-18 Poly unsaturated fatty acids (PUFA) and >20PUFA, (**B**) C_16_ fatty acids (**C**) C_18_ fatty acids excluding 18:5ω3 (**D**) Octadecapentaenoic acid (OPA), eicosapentaenoic acid (EPA), docosahexaenoic acid (DHA) and arachidonic acid ARA. Upper case letters are the identification references to the classes listed on the x-axes where specific species are listed in the Appendix A. Compilation from: [20,43,44,45,46,55,57,58,59,60,61,62,63,64,65,66,67,68,69,70,71,72,73,74,75,76,77,78,79,80,81,82,83,84,85,86,87,88]. Updated from [23].

**Figure 6 marinedrugs-17-00151-f006:**
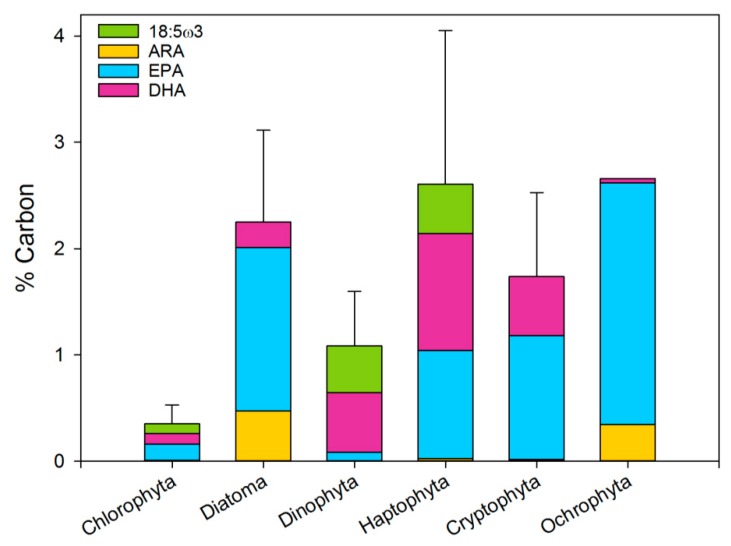
The proportion of Arachidonic acid (ARA, 20:4ω6), EPA, DHA and 18:5ω3 of the PUFA pool. Error bars are the sum of all four fatty acid standard errors.

**Table 1 marinedrugs-17-00151-t001:** Total average fatty acids (FA), polyunsaturated- and highly unsaturated fatty acids (PUFA and HUFA) as proportions of carbon content (% C ± SD), and ω3/ω6 and Eicosapentaenoic and docosahexaenoic acid ratio (EPA/DHA) (± standard deviation) of the 7 phytoplankton phyla.

Phyla	Total FA	PUFA	HUFA	HUFA + 18:5	ω3/ω6	DHA/EPA
**Chlorophyta**	8.6 ± 5.1	5.3 ± 3.6	0.3 ± 0.4	0.4 ± 0.5	8.1 ± 13.1	3.1 ± 4.8
**Diatoma**	13.6 ± 18.1	4.5 ± 6.0	2.3 ± 3.1	2.3 ± 3.1	15.2 ± 17.5	9.2 ± 9.0
**Dinophyta**	5.5 ± 2.2	1.6 ± 0.9	0.6 ± 0.5	1.1 ± 0.8	23.1 ± 15.8	0.3 ± 0.3
**Haptophyta**	14 ± 3.7	5.5 ± 2.2	2.5 ± 2.3	2.9 ± 2.0	5.0 ± 5.4	1.2 ± 3.2
**Cyanobacteria**	?	0	0	0	2.5 ± 2.3	-
**Cryptophyta**	9.4 ± 7.7	5.9 ± 5.3	1.8 ± 1.2	1.8 ± 1.2	7.0 ± 7.3	1.8 ± 1.1
**Ochrophyta**	13.1	4.5	3.9	3.9	5.4	0

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
