# Peer review of "Fatty Acid Profiles and Production in Marine Phytoplankton"

_marinedrugs, 2019, doi:10.3390/md17030151_

Round 1
Reviewer 1 Report
This paper of fatty acid profiles of marine phytoplankton provides a comprehensive review that will be of interest for a wide readership. It is generally well written and offers new information compared to other reviews of marine phytoplankton fatty acid profiles, but I have few concerns.
The first is related to selection of the profiles included in the meta-analysis. The author states that 160 profiles from 38 publications were selected, but does not offer any information on the selection process. Please clarify why these were selected and others where left out. Also, I find it odd that any statistical analysis of the data has not been done. The fatty acid profiles have been grouped but no information is provided on how it was done.
My second concern is the presentation of the results. I don’t like that in Figure 5 (and elsewhere) different 16:2 and 16:3 fatty acids are lumped together. From the figure a reader might get the impression that diatoms and chlorophytes have the same 16:2 and 16:3 fatty acids, which most likely is not the case. I understand that it might not be possible to include all the 16PUFAs separately in the figure but this should at least be discussed in the text.
Also, I would include the carbon concentrations of HUFA and PUFA in Table 1, and not in Figure 6. It would make the differences among HUFAs in the figure more clear. I feel that the amounts of fatty acids as fractions of carbon are what the other reviews of this topic have not addressed and these should be presented as clearly as possible. It is what this review offers compared to for example a recently published similar review by Cañavate (2018) that was based on 129 publications. Please site that review as well, and explain what new the current one offers.
I would appreciate if the full fatty acid profiles of marine microalgae were in a table format in the appendix. Including also the total concentration (as carbon fraction) where it is available. Giving the values (mean + variation) makes this data much more usable. The Table A1 is not clear. It might have been scrambled during the submission process since the column titles seem to be in wrong places. Please check this.
Minor comments:
Line 11: poly unsaturated = polyunsaturated
Line 14: reviles = reveals?
Figure 1. Explain EAA.
Figure 3. What about 22:5w6?
L130-131: The w-3 pathway does not start from 18:1w9, but with 18:2w6. It’s ok in the figure but please clarify the text.
L 213: Arachidonic acid misspelled here and elsewhere.
L304-311: Is this paragraph misplaced? It refers to “next chapter” but that does not exist. Also, I think this part merits a reference to Galloway and Winder (2015) who found that most of the variation in phytoplankton fatty acid profiles is related to phylogeny and not to environmental factors.
Author Response
First I would like to thank both reviewers for constructive review of the manuscript, and excellent suggestions to improve the review. My response is in bold after the reviewers comment
Reviewer 1.
This paper of fatty acid profiles of marine phytoplankton provides a comprehensive review that will be of interest for a wide readership. It is generally well written and offers new information compared to other reviews of marine phytoplankton fatty acid profiles, but I have few concerns.
The first is related to selection of the profiles included in the meta-analysis. The author states that 160 profiles from 38 publications were selected, but does not offer any information on the selection process. Please clarify why these were selected and others where left out. Also, I find it odd that any statistical analysis of the data has not been done. The fatty acid profiles have been grouped but no information is provided on how it was done.
- I have now included a short method section where the specifics of the selection are shortly outlined. I removed table A1 and added as a supplementary material all the raw data with the averaging and standard deviation of the profiles are presented after phyla, classes, and genera are given. Table 1 and Figure 6 have been improved giving standard deviations.
My second concern is the presentation of the results.
I don’t like that in Figure 5 (and elsewhere) different 16:2 and 16:3 fatty acids are lumped together. From the figure a reader might get the impression that diatoms and chlorophytes have the same 16:2 and 16:3 fatty acids, which most likely is not the case. I understand that it might not be possible to include all the 16PUFAs separately in the figure but this should at least be discussed in the text.
- I have now redrawn figure 5 and include the specifics of the 16 and 18 PUFA. The supplementary Excel sheet with all the profiles includes all the raw data used in the analysis to show the variation in the profiles that are hard to show on a composite figure. The text has been adjusted accordingly.
Also, I would include the carbon concentrations of HUFA and PUFA in Table 1, and not in Figure 6. It would make the differences among HUFAs in the figure more clear.
- An excellent suggestion that is now included.
I feel that the amounts of fatty acids as fractions of carbon are what the other reviews of this topic have not addressed and these should be presented as clearly as possible. It is what this review offers compared to for example a recently published similar review by Cañavate (2018) that was based on 129 publications. Please site that review as well, and explain what new the current one offers.
- I was not aware of this publication which is now included. I have put the carbon fraction as a specific sub section so it gets more focus in the review, and have elaborated the text.
I would appreciate if the full fatty acid profiles of marine microalgae were in a table format in the appendix. Including also the total concentration (as carbon fraction) where it is available. Giving the values (mean + variation) makes this data much more usable.
- As mentioned above, I am happy to offer — as supplementary material — a spread sheet with all the raw fatty acid profiles per species with the references to each. Those include the averages and standard deviations of phyla, genus and classes.
The Table A1 is not clear. It might have been scrambled during the submission process since the column titles seem to be in wrong places. Please check this.
- The table has been deleted as all the information is now available in the supplementary data file.
Minor comments:
Line 11: poly unsaturated = polyunsaturated - Corrected
Line 14: reviles = reveals? - Corrected
Figure 1. Explain EAA. - I redid Figure 1 and removed EAA but it is mentioned in the text. The removal is also to be consistent, as I did not specify EFA in the table. I removed WE - not very common in phytoplankton - and included GL (forgotten before). The figure text has been revised accordingly.
Figure 3. What about 22:5w6? - The elongation and desaturation path has been added into the figure and reference added accordingly.
L130-131: The w-3 pathway does not start from 18:1w9, but with 18:2w6. It’s ok in the figure but please clarify the text. - Thanks! Corrected
L 213: Arachidonic acid misspelled here and elsewhere. - Thanks! Corrected
L304-311: Is this paragraph misplaced? It refers to “next chapter” but that does not exist. - This was a bad copy editing from my part prior to submission.
Also, I think this part merits a reference to Galloway and Winder (2015) who found that most of the variation in phytoplankton fatty acid profiles is related to phylogeny and not to environmental factors. - This was mentioned, without references before, so now rewritten with reference to Galloway and Winder.
Reviewer 2 Report
Dear author your paper is a useful combilation of information especially for the industry that turns into the ocean for novel products. There are some minor mistakes mentioned on the text and some suggestions to improve the manuscript.

Author Response
First I would like to thank both reviewers for constructive review of the manuscript, and excellent suggestions to improve the review. My response is in bold after the reviewers comment
Reviewer 2
I copied all the comments the reviewer made in the manuscript file, referring to the pages the comments were made.
- Linguistic corrections have been taken care off on all pages.
Page 1: - References have been added to the zooplankton as trophic vectors.
Page 2: Subtitle head has been changed as suggested.
Page 3: Elaborate text on Protein and Carbohydrate
- I have added few lines on each macronutrient.
Page 3-4. Inclusion of more recent literature studies
- Thank you for pointing those out - I have included some of the suggested studies (Mühroth et al, Zulu et al . Harwood and Guchina.) - see next point.
Page 4: related to previous. the comments on acetyl-CoA - “acetyl-CoA may come from within the plastid or may come from cytosol” Krebs cycle - “this is part of the glycolytic pathway in the cytosol” and ER - “it depends whether the esterification is on phosphoglycerolipids (in the ER) or on galactoplipids in the plastid stroma. And there are the betain lipids (extra-plastidial)”
- I have included the information from the suggested papers and included in the text more detail on where the synthesis take place with appropriate citations.
Page 5:
Essential - “however, even in the studies you refer to, the ability of several animal groups in the aquatic to elongate and desaturate LA and ALA to LC-PUFA exists...”
-I thought I had this covered in the sentence, but due to this comment the sentence has been reworded
References in figure legend
- The reference program played some tricks on me. Now all references are checked and some additional references have been added.
glycerol storage (hypersaline Dunaliella).
- The sentence has been removed - thanks for pointing this out - something that is actually very clear in the paper I cited - but I missed.
Page 9:
cost - “most studies on absolute values (mg/g or pg/cell) show accumulation of lipids in neutral forms, while polar lipids are constant in the membranes (percentages of course change in the way stated). “
- I removed the “cost of PUFA and PL”
direct correlation - “there is though an effect of light changes affecting the size of thylakoids mostly composed by galactolipids with high amounts of ALA ...”
-I searched and found the study of Guiheneuf et al and could incorporate this point into the paragraph
decrease- “see previous comments...of course when nutrient limitation becomes critical it affects the size of thylakoids and other cell membranes decreasing thus the absolute amounts of PUFA. In general N limitation shifts synthetic pathways to non nitrogenous compounds (carbohydrates or lipids, seldom both) until limitation becomes critical affecting the turnover of enzymes and ability to repair or synthesize membranes hence, recycling PL and GL and associated PUFA. “
- Thank you - I allowed myself to weave this comment in the paragraph.
lipid synthesis - “or carbohydrate for some groups...” Incorporated into the reshuffled text
decrease - “which is logic as the latter do not store C in lipids but in carbohydrates and P limitation affects their ability to synthesize phosphoglycerolipids...” Nice comment which is included
lowered “when really and not apparently lowered...” - text changed
phytoplankton - “please try not to treat phytoplankton as a group of similars... As you clearly mentioned from the begining, it comprises at least seven phyla i.e. very high ranking taxa exhibiting fundamental diferences. Only photosynthesis is common (to some extent)... cyanobacteria are indeed bacteria and chlorophytes are primordial plants...” - I went through the MS with this in mind and corrected where I felt it was appropriate.
Page 10:
- high economic value - “it could be useful to add some references here...” - few references have been added
- “please use the modern term cyanobacteria...” -Corrected
- Highly seasonal - This has been clarified.
- Litterature in Table A1. I have removed the table, and included a supplementary file with all the raw fatty acid profiles used in the analysis. In that file all the references are included.
Round 2
Reviewer 1 Report
I am very pleased with the edits made by the author, and I think the manuscript is much improved. I only have few minor comments:
Line 19: What about cyanobacteria?
Line 270 and figure 5: Please double-check which 16PUFAs diatoms have. I have never heard of them having 16:3w3 or 16:4w3. Neither did Canavate (2018) report any w-3 16PUFAs in diatoms.
Line 287: Eustigmatophyceae
Author Response
Thank you again for a thorough review.
Line 19: What about cyanobacteria? - Yes, thanks, this is added.
Line 270 and figure 5: Please double-check which 16PUFAs diatoms have. I have never heard of them having 16:3w3 or 16:4w3. Neither did Canavate (2018) report any w-3 16PUFAs in diatoms. For 16:3 is presented as 16:3n1 or 3, so either a 16:3n1 or 16:3n3 - I went to the original tables and see that for diatoms it is 16:3n1 - but 16:3n3 for chlorophyta. I do now separate those in figure 5 and the supplementary table.
For 16:4n3. I am very grateful that you commented on this. Most of the 16:4n3 fatty acid data on diatoms are from my own analyses. Therefore I could go to the original data and saw that they are were analysed as 16:4 not 16:4n3. (At that time I did not have the 16:4n3 or 16:4n1 standards). This is therefore a mistake that happened somewhere along the line while working with the original literature compilation where I had all the phyla on one page in the spread sheet - and at some later time decide to split 16:4 into n1 and n3. I made new Figure 5 where 16:4n1 are now presented as 16:4n1 and 16:4 where I add my data. The data from Viso and Marty give 16:4n3 in their diatom analyses and those are still presented in the averages - a small proportion.
Line 287: Eustigmatophyceae - I am not sure what this refers to, but I have changed the phylum to Ochrophyta (new in Algabase since I did the compilation) Class Eustigmatophyceae and changed it accordingly in the text all figures and tables.
Additional changes
An unclear sentence in the abstract was rewritten.